# Attention-based Generative Latent Replay: A Continual Learning Approach for WSI Analysis

**Pratibha Kumari**[1*]          Pratibha.Kumari@ur.de
**Daniel Reisenbüchler**[1*]          Daniel.Reisenbuechler@ur.de
**Afshin Bozorgpour**[1]          Afshin.Bozorgpour@ur.de
**Nadine S. Schaadt**[2]          Schaadt.Nadine@mh-hannover.de
**Friedrich Feuerhake**[2]          Feuerhake.Friedrich@mh-hannover.de
**Dorit Merhof**[1,3]          Dorit.Merhof@ur.de

[1] *Faculty of Informatics and Data Science, University of Regensburg, Germany*

[2] *Institute of Pathology, Hannover Medical School, Hannover, Germany*

[3] *Fraunhofer Institute for Digital Medicine MEVIS, Bremen, Germany*

## Abstract

Whole slide image (WSI) classification has emerged as a powerful tool in computational pathology, but remains constrained by domain shifts, e.g., due to different organs, diseases, or institution-specific variations. To address this challenge, we propose an **A**ttention-based **G**enerative **L**atent **R**eplay **C**ontinual **L**earning framework (AGLR-CL), in a multiple instance learning (MIL) setup for *domain incremental* WSI classification. Our method employs Gaussian Mixture Models (GMMs) to synthesize WSI representations and patch count distributions, preserving knowledge of past domains without explicitly storing original data. A novel attention-based filtering step focuses on the most salient patch embeddings, ensuring high-quality synthetic samples. This privacy-aware strategy obviates the need for replay buffers and outperforms other buffer-free counterparts while matching the performance of buffer-based solutions. We validate AGLR-CL on clinically relevant biomarker detection and molecular status prediction across multiple public datasets with diverse centers, organs, and patient cohorts. Experimental results confirm its ability to retain prior knowledge and adapt to new domains, offering an effective, privacy-preserving avenue for domain incremental continual learning in WSI classification.

**Keywords:** Whole Slide Image Analysis, Computation Pathology, Biomarker Screening, Continual Learning

## 1 Introduction

Recent advances in computational pathology (CPath) and digitizing WSIs have transformed histopathology image analysis, driving significant progress in automated disease detection and biomarker assessment. However, WSI classification remains challenging due to the gigapixel resolution and the lack of pixel-level annotations. A common strategy divides WSIs into manageable patches, which are processed offline by vision encoding models to obtain feature sequences. Notably, self-supervised pretraining has enabled the development of domain-specific foundation models (FMs) that outperform out-of-domain counterparts (Xu

---

∗. These authors contributed equally to this work.

et al., 2024; Chen et al., 2024; Wang et al., 2023), such as ImageNet-pretrained models. The conversion of patch-level features into slide-level predictions is achieved through MIL by aggregation of these features.

Despite these advancements, WSI classification models still face challenges in clinical settings. Variability in morphological features, originating from differences in organ-specific biology, staining protocols, scanner manufacturers, and patient cohorts, induces distribution shifts that degrade performance on new datasets. Conventional MIL models struggle to generalize when WSIs are acquired from diverse hospitals and settings. Fine-tuning on new datasets is a common adaptation strategy; however, it often leads to catastrophic forgetting (CF) (Kirkpatrick et al., 2017; Kumari et al., 2024a). On the other hand, continual learning (CL) has emerged as a promising solution for evolving medical data while mitigating CF (Sadegheih et al., 2025; Kumari et al., 2025b,a). By enabling continuous knowledge accumulation, CL enhances model robustness and adaptability in clinical settings and facilitates forward knowledge transfer, e.g., from frequently stained datasets in H&E or PAS to those for follow-up diagnostics like CD8 or TRI (Kumari et al., 2024b). Although buffer-based methods, which store selected past samples, typically yield superior performance (Derakhshani et al., 2022; Bhatt et al., 2024), their applicability to WSIs is hindered by storage and privacy constraints. Existing WSI CL research is limited, focusing primarily on buffer-based and class incremental methods (Huang et al., 2023; Zhu et al., 2024). To address these limitations, we propose AGLR-CL, a buffer-free generative replay approach for domain incremental WSI classification. AGLR-CL models past domain distributions with GMMs trained on patch embeddings and counts. For each domain, class-wise multivariate GMMs and one-dimensional GMMs capture prior data distribution. In subsequent domains, synthetic data sampled from these GMMs are combined with new WSIs to update the MIL model, thus avoiding real data storage and preserving privacy. We validate AGLR-CL on multiple tasks across domain incremental datastes including various centers and organs. Extensive experiments show that AGLR-CL effectively retains prior knowledge and adapts to new domains, surpassing other buffer-free methods and achieving performance close to buffer-based methods. Our main contributions are:

**(1) Domain incremental CL for MIL.** To our knowledge, we introduce domain incremental CL for MIL for the first time and present a GMM and attention-based filtering for effective re-sampling of past data across domains.

**(2) Broad applicability and increased privacy.** Across CPath tasks, including biomarker screening and molecular status predictions, our AGLR-CL consistently surpasses buffer-free methods and is on par with buffer-based methods, while avoiding WSI storage and thus increasing privacy.

## 2 Method

A flowchart of the proposed approach is shown in Fig. 1. In the following, we detail MIL-based WSI classification, CL settings, and our AGLR-CL framework, which incorporates an attention-based selection for GMM training and synthetic embedding generation for a latent replay mechanism.

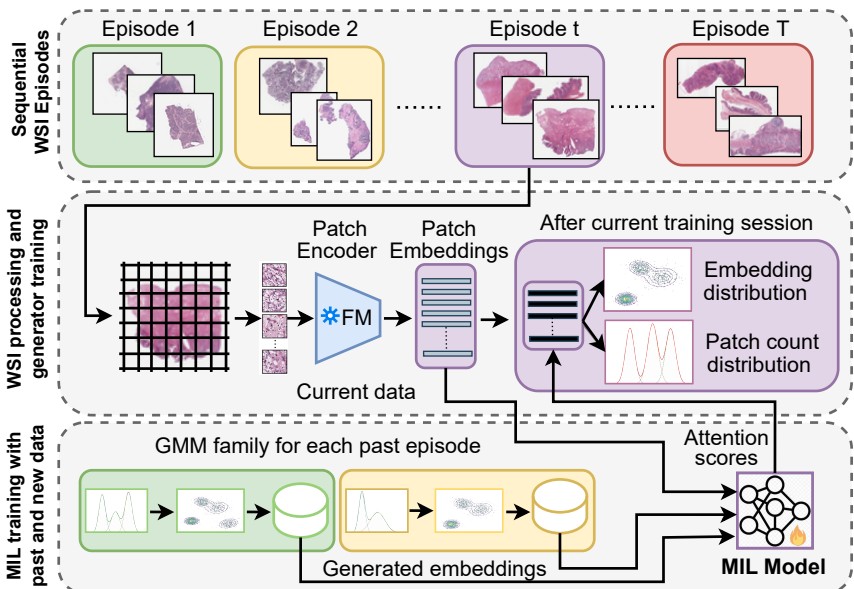

Figure 1: **Privacy-aware domain incremental AGLR-CL framework.** Following WSI tessellation, a frozen FM-based encoder generates a sequence of tractable embeddings. A MIL aggregator is then trained on the current episode, with high-attention features selected to fit GMMs for patch embeddings and counts. In subsequent episodes, historical data is revisited by re-sampling synthetic WSIs using the per-episode GMMs.

## 2.1 MIL-based WSI Classification

We adopt a standard preprocessing pipeline, partitioning each WSI into $n$ non-overlapping patches $p_i \in \mathbb{R}^{512 \times 512 \times 3}$. A pretrained CPath FM is then used to extract patch embeddings, resulting in a feature sequence $\{f_i\}_{i=1}^{n} \in \mathbb{R}^{n \times D}$, with $D$ denoting the latent dimension. These embeddings are aggregated using a learnable MIL model $\mathcal{M}$. Specifically, we employ AB-MIL (Ilse et al., 2018), which embeds each feature into a lower-dimensional space $d$ via a linear layer and applies an attention mechanism to assign instance-level weights. The weighted embeddings are summed and fed into a classification head for WSI prediction.

## 2.2 Continual Learning Configuration

We consider a CL pipeline for WSI classification, where datasets arrive sequentially (defined as episodes), $\{\mathcal{D}_1, \mathcal{D}_2, \dots, \mathcal{D}_T\}$, each representing a distinct domain $t \in \{1, ..., T\}$. At training domain $t$, the model $\mathcal{M}$ has access to the current dataset $\mathcal{D}_t$ only, while evaluation is performed on test sets from all episodes $\{\mathcal{D}_1, \mathcal{D}_2, \dots, \mathcal{D}_T\}$. Unlike approaches that retain a buffer of past samples (Zhu et al., 2024; Huang et al., 2023), our method addresses CF while avoiding WSI storage, a critical requirement in privacy-sensitive domains.

## 2.3 GMM-based Synthetic Embedding Generation

WSIs inherently contain a variable number of patches. To generate synthetic WSI representations, we model both the patch counts and patch embeddings using GMMs. For a new dataset $\mathcal{D}_t$, we estimate class-specific multivariate models $\text{GMM}_{\text{emb}}^t$ on the collected patch embeddings to capture feature variations. Concurrently, one-dimensional models $\text{GMM}_{\text{count}}^t$ are fitted on patch counts to account for tissue variability across WSIs. Since not all patches contribute meaningfully to the classification task, we introduce an attention-guided filtering step prior to GMM estimation. After the $t^{\text{th}}$ training session with classifier $\mathcal{M}_t$, attention scores are computed for patches across each WSI in the dataset of the current episode $\mathcal{D}_t$. Low-attention patches are discarded, retaining only the top $q\%$ for subsequent processing. Consequently, the feature sequence $\{f_i\}_{i=1}^{n_j}$ of each WSI $\mathcal{W}_j \in \mathcal{D}_t$ with $j \in \{1, \ldots, |\mathcal{D}_t|\}$ is reduced to $m_j < n_j$. Next, we define GMMs for WSI embedding generation. The probability density function for a feature embedding is its likelihood under a $K$-component GMM, given by

$$p(f_i|\Theta) = \sum_{k=1}^{K} \pi_k \mathcal{N}(f_i|\mu_k, \Sigma_k), \tag{1}$$

where $\mathcal{N}(f_i|\mu_k, \Sigma_k)$ represents a Gaussian density function with $k$ mixture parameters given by mean $\mu_k$ and covariance $\Sigma_k$, which are defined by

$$\mu_k = \frac{\sum_{i=1}^{n} \gamma_{ik} f_i}{\sum_{i=1}^{n} \gamma_{ik}}, \quad \Sigma_k = \frac{\sum_{i=1}^{n} \gamma_{ik}(f_i - \mu_k)(f_i - \mu_k)^T}{\sum_{i=1}^{n} \gamma_{ik}}, \tag{2}$$

with the responsibility $\gamma_{ik}$ computed via the Expectation-Maximization (Dempster et al., 1977) to update the parameters $\mu_k, \Sigma_k$ iteratively:

$$\gamma_{ik} = \frac{\pi_k \mathcal{N}(f_i|\mu_k, \Sigma_k)}{\sum_{j=1}^{K} \pi_j \mathcal{N}(f_i|\mu_j, \Sigma_j)}. \tag{3}$$

The optimal $K$ is selected by minimizing the Bayesian Information Criterion (BIC) (Fraley and Raftery, 1998) over candidate values. The estimated parameters define a generative model that facilitates on-the-fly sampling of synthetic patch embeddings mimicking $\mathcal{D}_t$. Concurrently, the number of patches $\hat{n}_j$ in a synthetic WSI $\mathcal{W}_j \in \mathcal{D}_t$ is determined by sampling from $\text{GMM}_{\text{count}}^t$, ensuring that the generated WSIs have a realistic patch count, or in other words, tissue variability. We denote the union of GMMs created for each episode as GMM family.

## 2.4 Generative Latent Replay

During the $t^{\text{th}}$ training session, the current dataset $\mathcal{D}_t$ is expanded with synthetic WSIs generated from the $t-1$ GMM families learned in all previous episodic datasets $\{\mathcal{D}_1, \mathcal{D}_2, \ldots, \mathcal{D}_{t-1}\}$. For each past session $t' < t$, synthetic WSI embeddings are generated as feature sequences $\{f_i\}_{i=1}^{\hat{n}_j}$ for $j \in \{1, \ldots, |\mathcal{D}_{t'}|\}$. To this end, we first sample a patch count $\hat{n}_j$ from $\text{GMM}_{\text{count}}^{t'}$ and subsequentially drawing $\hat{n}_j$ patch embeddings from $\text{GMM}_{\text{emb}}^{t'}$. The number of synthetic WSIs matches the WSIs count in $\mathcal{D}_t$ while preserving the class ratio previously observed in

Table 1: **Dataset statistics.** Overview of data cohorts across organs, centers, and tasks, to create both homogeneous and heterogeneous domain shifts. We used a patient-stratified split into Train/Test sets to avoid data leakage from individual datasets.

| | Name | Train Class 0/1 | Test Class 0/1 | Organ | Center |
|---|---|---|---|---|---|
| **MSI** | TCGA-CRC | 303/52 | 79/13 | Colorectal | multiple |
| | CPTAC-COAD | 138/41 | 30/12 | Colon | $C1$ |
| | PAIP-CRC | 28/9 | 7/3 | Colorectal | $C2$ |
| | TCGA-STAD | 239/48 | 62/12 | Stomach | multiple |
| | TCGA-UCEC | 340/92 | 88/25 | Uterine | multiple |
| **TMB** | TCGA-STAD | 261/66 | 67/16 | Stomach | multiple |
| | TCGA-UCEC | 278/152 | 68/38 | Uterine | multiple |
| | TCGA-NSCLC | 533/280 | 143/67 | Lung | multiple |
| | TCGA-CRC | 350/65 | 90/16 | Colorectal | multiple |
| | TCGA-BRCA | 853/26 | 210/6 | Breast | multiple |
| **HER2** | TCGA-BRCA | 469/135 | 114/32 | Breast | multiple |
| | CPTAC-BRCA | 266/38 | 56/7 | Breast | $C4$ |
| | BCNB | 625/221 | 156/56 | Breast | $C5$ |
| **PR** | TCGA-BRCA | 275/577 | 71/145 | Breast | multiple |
| | CPTAC-BRCA | 114/159 | 41/34 | Breast | $C4$ |
| | BCNB | 214/632 | 54/158 | Breast | $C5$ |

Table 2: **Dataset episodes detail.** We curate heterogeneous organ/center (a2, a1, a5) and homogeneous center (a3, a4) shifts to obtain domain incremental episodes.

| Seq. | Task | Dataset episodes |
|---|---|---|
| a1 | MSI | TCGA-STAD → PAIP-CRC → TCGA-UCEC |
| a2 | MSI | PAIP-CRC → TCGA-CRC → TCGA-STAD → CPTAC-COAD → TCGA-UCEC |
| a3 | PR | TCGA-BRCA → CPTAC-BRCA → BCNB |
| a4 | HER2 | TCGA-BRCA → BCNB → CPTAC-BRCA |
| a5 | TMB | TCGA-STAD → TCGA-UCEC → TCGA-NSCLC → TCGA-CRC → TCGA-BRCA |

$\mathcal{D}_{t'}$. These synthetic samples are then combined with the real WSIs from the current session to form a hybrid training set. By integrating synthetic data through generative replay, the model continuously reinforces knowledge from previous episodes, thereby mitigating CF, reducing overfitting to new data, and eliminating the need to store real historical samples.

## 3 Experiments

**Datasets.** We consider multiple publicly available WSI datasets for biomarker screening of microsatellite instability (MSI) and tumor mutational burden (TMB), binarizing TMB numeric values at 10 mutations/megabase. We also perform molecular status prediction of progesterone receptor (PR) and human epidermal growth factor receptor 2 (HER2) in breast cancer. Specifically, we explore data repositories such as The Cancer Genome Atlas (TCGA) (Cancer Genome Atlas Network, 2012), Clinical Proteomic Tumor Analysis Consortium (CPTAC) (Edwards et al., 2015), PAIP2020 challenge (Kim et al., 2023), and

Table 3: **Performance comparison across CL methods.** Past raw data (PRD) marks if past WSIs need to be stored. Red and blue indicate first and second best performances by all CL methods. Underline shows best performance in buffer-free CL.

| Task (Seq.) | PRD | Method | weighted F1 | | | AUROC | | | AUPRC | | |
|---|---|---|---|---|---|---|---|---|---|---|---|
| | | | ACC | A | BWT | ACC | A | BWT | ACC | A | BWT |
| MSI (a1) | ✗ | Naive | 65.20 | 65.45 | -17.32 | 61.17 | 66.7 | -23.67 | 32.31 | 39.93 | -25.97 |
| | ✓ | Joint | 69.84 | — | — | 65.96 | — | — | 41.21 | — | — |
| | ✓ | Cumulative | 76.02 | 76.71 | -0.41 | 77.63 | 81.6 | -6.13 | 54.85 | 59.23 | -7.67 |
| | ✓ | Replay | 73.02 | 75.07 | -2.84 | 67.87 | 73.22 | -5.27 | 47.88 | 50.44 | -5.51 |
| | ✓ | GDumb | 64.52 | 65.34 | -0.5 | 58.54 | 61.91 | -0.19 | 34.96 | 36.83 | -5.55 |
| | ✗ | LwF | 59.94 | 64.53 | -11.72 | 53.42 | 59.95 | -31.47 | 31.68 | 40.22 | -23.5 |
| | ✗ | EWC | 64.17 | 70.06 | -21.79 | 55.16 | 65.74 | -33.83 | 37.27 | 46.57 | -27.65 |
| | ✗ | SI | 66.25 | 68.15 | -16.64 | 60.26 | 65.77 | -37.24 | 45.24 | 51.74 | -23.68 |
| | ✗ | EVCL | 69.09 | 69.38 | -13.97 | 59.29 | 65.26 | -36.94 | 38.22 | 48.23 | -27.76 |
| | ✗ | **Ours** | 69.91 | 73.84 | -4.45 | 64.88 | 72.87 | -20.18 | 38.38 | 49.44 | -23.84 |
| MSI (a1) | ✗ | Naive | 68.19 | 74.59 | -2.63 | 70.82 | 76.33 | 1.27 | 45.44 | 54.46 | 6.78 |
| | ✓ | Joint | 79.95 | - | - | 74.07 | - | - | 56.87 | - | - |
| | ✓ | Cumulative | 72.90 | 76.51 | 3.36 | 72.13 | 76.53 | 4.03 | 41.11 | 50.8 | 2.92 |
| | ✓ | Replay | 68.34 | 72.35 | -2.59 | 68.32 | 75.01 | -1.6 | 40.51 | 47.24 | -3.41 |
| | ✓ | GDumb | 71.66 | 68.05 | 16.11 | 60.69 | 60.42 | 3.39 | 40.47 | 38.56 | 2.19 |
| | ✗ | LwF | 68.71 | 72.68 | -3.86 | 62.51 | 70.76 | -5.0 | 34.76 | 42.47 | -6.39 |
| | ✗ | EWC | 65.03 | 70.49 | -7.01 | 59.17 | 69.81 | -9.16 | 37.52 | 45.9 | -1.26 |
| | ✗ | SI | 68.58 | 71.53 | -7.1 | 63.51 | 70.68 | -8.5 | 36.97 | 47.26 | -5.66 |
| | ✗ | EVCL | 67.16 | 71.05 | -5.79 | 59.85 | 67.71 | -11.04 | 34.47 | 44.81 | -3.14 |
| | ✗ | **Ours** | 78.01 | 74.1 | -1.89 | 69.11 | 74.05 | -2.32 | 47.66 | 52.94 | 9.61 |
| PR (a3) | ✗ | Naive | 64.29 | 67.29 | -7.44 | 66.85 | 69.08 | -9.85 | 73.34 | 74.06 | -8.04 |
| | ✓ | Joint | 69.76 | — | — | 71.84 | — | — | 77.69 | — | — |
| | ✓ | Cumulative | 67.60 | 66.73 | 0.37 | 71.40 | 72.0 | -0.5 | 77.73 | 77.31 | -0.48 |
| | ✓ | Replay | 70.77 | 67.76 | 3.54 | 68.48 | 69.18 | -4.56 | 74.96 | 74.31 | -2.46 |
| | ✓ | GDumb | 67.21 | 63.22 | 11.52 | 72.82 | 67.97 | 9.7 | 77.51 | 74.98 | 2.5 |
| | ✗ | LwF | 66.91 | 64.1 | -2.05 | 70.35 | 68.16 | -1.79 | 76.00 | 73.28 | -0.65 |
| | ✗ | EWC | 63.28 | 64.27 | -4.82 | 66.30 | 66.87 | -7.13 | 73.36 | 72.13 | -5.03 |
| | ✗ | SI | 63.55 | 65.03 | -6.61 | 66.77 | 68.28 | -6.64 | 74.72 | 74.34 | -3.42 |
| | ✗ | EVCL | 65.19 | 65.49 | -4.11 | 71.22 | 71.16 | -4.90 | 76.49 | 75.79 | -3.95 |
| | ✗ | **Ours** | 67.97 | 66.33 | -1.48 | 70.34 | 71.16 | -8.0 | 76.35 | 76.23 | -6.28 |
| HER2 (a4) | ✗ | Naive | 71.80 | 71.99 | -3.21 | 59.57 | 63.6 | -5.41 | 29.36 | 35.53 | -5.98 |
| | ✓ | Joint | 75.85 | — | — | 62.90 | — | — | 36.44 | — | — |
| | ✓ | Cumulative | 75.59 | 75.66 | 0.84 | 61.82 | 66.07 | 1.96 | 36.91 | 41.5 | 5.1 |
| | ✓ | Replay | 75.32 | 74.12 | 1.7 | 59.24 | 63.93 | 1.59 | 31.95 | 38.0 | 1.62 |
| | ✓ | GDumb | 75.64 | 73.43 | 3.56 | 62.13 | 64.77 | 8.47 | 32.95 | 37.69 | 7.5 |
| | ✗ | LwF | 72.29 | 72.16 | -4.59 | 58.12 | 62.71 | -5.04 | 28.07 | 34.21 | -5.44 |
| | ✗ | EWC | 71.44 | 72.31 | -5.39 | 61.22 | 64.97 | -2.85 | 28.90 | 35.74 | -5.34 |
| | ✗ | SI | 71.62 | 72.74 | -3.39 | 55.37 | 62.21 | -0.53 | 28.27 | 35.92 | -1.58 |
| | ✗ | EVCL | 71.26 | 72.85 | -3.44 | 54.68 | 61.46 | -8.96 | 24.71 | 33.61 | -10.99 |
| | ✗ | **Ours** | 76.94 | 75.11 | -0.78 | 66.18 | 66.79 | -0.49 | 35.07 | 37.99 | -1.97 |
| TMB (a5) | ✗ | Naive | 70.78 | 68.84 | -7.16 | 49.50 | 59.27 | -11.33 | 22.84 | 35.24 | -7.17 |
| | ✓ | Joint | 74.35 | — | — | 61.48 | — | — | 34.49 | — | — |
| | ✓ | Cumulative | 74.48 | 69.75 | 0.51 | 55.40 | 60.4 | -0.03 | 32.91 | 37.88 | 1.95 |
| | ✓ | Replay | 71.62 | 69.4 | -2.09 | 50.77 | 60.39 | -7.39 | 26.53 | 36.84 | -4.44 |
| | ✓ | GDumb | 72.98 | 67.89 | 0.49 | 54.72 | 55.47 | 4.76 | 28.08 | 33.37 | 3.07 |
| | ✗ | LwF | 70.32 | 69.05 | -4.42 | 55.51 | 60.58 | -6.12 | 28.35 | 37.93 | -5.11 |
| | ✗ | EWC | 70.63 | 67.32 | -11.63 | 48.34 | 59.13 | -15.18 | 20.60 | 33.09 | -13.94 |
| | ✗ | SI | 70.44 | 67.76 | -5.29 | 43.98 | 56.4 | -13.12 | 20.33 | 34.53 | -7.99 |
| | ✗ | EVCL | 70.46 | 69.26 | -4.15 | 50.08 | 61.96 | -7.85 | 21.72 | 37.85 | -4.65 |
| | ✗ | **Ours** | 73.04 | 69.58 | -4.73 | 57.45 | 62.39 | -10.6 | 31.53 | 38.53 | -5.25 |

Early Breast Cancer Core-Needle Biopsy WSI (BCNB) (Xu et al., 2021). For TCGA and CPTAC, we obtain labels from cbioportal.org. Table 1 summarizes the datasets, detailing tasks, train/test volume, organs, and centers. Datasets from multiple centers are marked *multiple*; otherwise, labeled as $(C1, C2, \dots)$.

**Continual Learning Episodes.** To comprehensively evaluate AGLR-CL, we create multiple sequences from datasets listed in Table 1, each having several datasets as episodes. WSI datasets in each sequence exhibit differences in terms of organ, center, and mixed shifts to create domain incremental scenarios within the CL framework, as detailed in Table 2.

**Continual Learning Baselines.** We compare our method against various CL baselines, including regularization methods such as EWC (Kirkpatrick et al., 2017), SI (Zenke et al., 2017), LwF (Li and Hoiem, 2018), and EVCL (Batra and Clark, 2024) and rehearsal methods such as GDumb (Prabhu et al., 2020) and Replay (Rolnick et al., 2019). We report lower

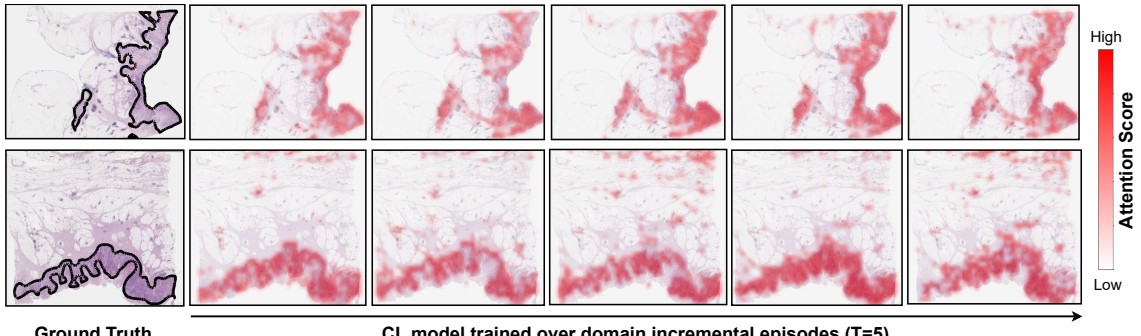

**Ground Truth**    **CL model trained over domain incremental episodes (T=5)**

Figure 2: **Attention heatmaps for AGLR-CL across domain shifts in MSI prediction.** Attentions scores for WSIs from $\mathcal{D}_1$ (PAIP-CRC) by the model trained over five CL stages in a2 reflect that past knowledge acquired from $\mathcal{D}_1$ is preserved.

bound performance by *naive* approach and upper bound performances by *joint* and *cumulative* approaches. *Naive* corresponds to traditional fine-tuning with only current dataset, *joint* uses all datasets simultaneously, and *cumulative* sequentially incorporates seen datasets.

**Implementation Details.** We extracted patches using the CLAM library (Lu et al., 2021) and employed the pre-trained UNI pathology FM (Chen et al., 2024) for feature extraction across all methods. Data leakage is avoided as considered datasets (Table 1) for our CL experiments are disjoint from UNI's training data. The buffer for Replay and GDumb is set to 100. For SI, EWC, EVCL, and LwF, the regularizing factor was set to 1 by following the literature (Kumari et al., 2024b). In our AGLR-CL, we keep $q$ as 80%. We select $K$ from $\{8, 16, 24\}$ for $\text{GMM}_{\text{emb}}^t$ and $\{1, 2, 3, 4, 5\}$ for $\text{GMM}_{\text{count}}^t$. To accommodate for class imbalances, we track weighted F1 score, AUROC, and AUPRC metrics. For sequential training and evaluation in CL with $T$ episodes, we consider $R \in \mathbb{R}^{T \times T}$ as train-test matrix (Kumari et al., 2025b) where cell $R_{ij}$ denote performance on $j^{th}$ datasets after $i^{th}$ training session with $\mathcal{D}_i$. We compute CL metrics from this matrix, including forgetting measure $BWT$ (Díaz-Rodríguez et al., 2018) and average performance on seen datasets using $ACC$ (Lopez-Paz and Ranzato, 2017), computed at the last episode and $A$ (Díaz-Rodríguez et al., 2018), computed at each training step. The CL metric $A$ captures both transfer learning and backward transfer capabilities (Özgün et al., 2020). The metrics are described in detail in the supplementary file. The larger these metrics, the better the performance. All experiments were run on a single NVIDIA H100 GPU.

## 4 Results

Table 3 compares AGLR-CL against competing approaches on MSI prediction, PR status, HER2 status, and TMB mutation. We report $ACC$, $A$, and $BWT$ based on weighted F1, AUROC, and AUPRC. Across sequences a1–a5, the naive update of the model exhibits lower performance ($ACC$ and $A$) and higher forgetting ($BWT$) compared to the cumulative, while joint training on all data provides an upper bound. Among CL methods, buffer-based methods (Replay and GDumb) generally achieve the best performance (red), with our approach

Table 4: **Ablation study for attention-based filtering (ABF).** Best in **bold**.

| Task | Seq. | w/o ABF | weighted F1 | | | AUROC | | | AUPRC | | |
|------|------|---------|------|------|------|------|------|------|------|------|------|
| | | | *ACC* | *A* | *BWT* | *ACC* | *A* | *BWT* | *ACC* | *A* | *BWT* |
| MSI | a1 | ✗ | 67.13 | 69.15 | -13.8 | 63.09 | 68.51 | -24.22 | **41.61** | 45.44 | -24.21 |
| | | ✓ | **69.91** | **73.84** | **-4.45** | **64.88** | **72.87** | **-20.18** | 38.38 | **49.44** | **-23.84** |
| MSI | a2 | ✗ | 66.16 | 67.75 | -9.84 | 68.59 | 73.30 | **-1.22** | 45.57 | 49.27 | 8.17 |
| | | ✓ | **78.01** | **74.10** | **-1.89** | **69.11** | **74.05** | -2.32 | **47.66** | **52.94** | **9.61** |
| PR | a3 | ✗ | 64.32 | 63.72 | -3.14 | 62.93 | 65.00 | **-6.92** | 69.85 | 70.90 | -8.5 |
| | | ✓ | **67.97** | **66.33** | **-1.48** | **70.34** | **71.16** | -8.0 | **76.35** | **76.23** | **-6.28** |
| HER2 | a4 | ✗ | 73.12 | 73.41 | -5.73 | 61.75 | 65.06 | -5.65 | 29.32 | 36.07 | -7.79 |
| | | ✓ | **76.94** | **75.11** | **-0.78** | **66.18** | **66.79** | **-0.49** | **35.07** | **37.99** | **-1.97** |
| TMB | a5 | ✗ | **73.58** | **70.32** | **-1.88** | **58.60** | **65.39** | **-2.95** | 30.98 | **38.86** | **-0.09** |
| | | ✓ | 73.04 | 69.58 | -4.73 | 57.45 | 62.39 | -10.6 | **31.53** | 38.53 | -5.25 |

following in a1 and a3 and surpassing them in a2, a4 and a5. Notably, when considering buffer-free methods only, our approach mostly delivers the best results (underlined). Thus, while slightly trailing buffer-based methods, our buffer-free solution offers a competitive alternative in privacy-sensitive applications.

Fig. 2 shows attention heatmaps for two WSIs from $\mathcal{D}_1$ (PAIP-CRC) in sequence a2 by model $\mathcal{M}$ over five training sessions, corresponding to sequential training with different datasets. It can be observed that high attention scores cover the annotated region in all CL training sessions. Interestingly, an organ-shift ($t = 3, 5$) creates a few artifacts compared to center-only shifts ($t = 2, 4$). Overall, consistent attention to the ground-truth area reflects that past knowledge is preserved while learning on new datasets with differences in centers and organs.

**Ablation.** Table 4 presents an ablation study on attention-based filtering for GMMs training. The results show that, except for sequence a5, GMMs trained with filtered patches consistently outperform those trained on all patch embeddings. For sequence a5, the slight drop may occur due to the high variability of morphological alterations in different TMB outcomes across organs, thus discarding certain patches can hurt data recovery.

## 5 Conclusion

We proposed AGLR-CL, a buffer-free generative latent replay framework enabling privacy-aware CL for WSI tasks including biomarker screening and molecular status predictions. Instead of maintaining a buffer, AGLR-CL leverages GMMs to synthesize past feature distributions, allowing the model to retain knowledge while adapting to new domains. Results demonstrate that AGLR-CL mitigates CF and achieves state-of-the-art performance in privacy-sensitive CL.

## Acknowledgments and Disclosure of Funding

This work was supported by the German Research Foundation (Deutsche Forschungsgemeinschaft, DFG) under project number 445703531. The authors gratefully acknowledge the computational and data resources provided by the Leibniz Supercomputing Centre (www.lrz.de).

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

# Supplementary Material

## Continual Learning Metrics

**ACC**: *ACC* represent average accuracy across all tasks, computed after learning with the last dataset (Lopez-Paz and Ranzato, 2017) computed as below

$$\frac{1}{T} \sum_{j=1}^{T} R_{T,j} \tag{4}$$

**A**: The CL metric $A$ (Díaz-Rodríguez et al., 2018) captures both transfer learning and backward transfer capabilities (Özgün et al., 2020). It shows the incremental learning ability of the model in a non-stationary environment which is computed as the average of entries in the lower-triangular matrix $(R)$ including the diagonal entries as follows:

$$\frac{2}{T(T+1)} \sum_{j \leq i} R_{i,j} \tag{5}$$

**BWT**: It shows the effect of learning a new dataset on previously already learned datasets. It is a way to measure stability, i.e., how well the model would retain the performance on previously acquired knowledge to prevent catastrophic forgetting. The larger the negative value of BWT, the larger the forgetting. Specifically, we compute BWT following (Díaz-Rodríguez et al., 2018) as shown below:

$$\frac{1}{T-1} \sum_{j=1}^{T-1} \frac{1}{\left| \{t_i\}_{i>j} \right|} \sum_{i>j} [R_{i,j} - R_{j,j}] \tag{6}$$

