# OpenReview forum: "Attention-based Generative Latent Replay: A Continual Learning Approach for WSI Analysis"
_MICCAI.org/2025/Workshop/COMPAYL — COMPAYL 2025_

### Official Review · Reviewer_vFgm · 2025-07-13
**Robust and Innovative Approach to Continual Learning on MIL**

**Rating:** 5
**Confidence:** 5

**Review:**

**Short Summary**

This paper introduces the Attention-based Generative Latent Replay Continual Learning (AGLR-CL) framework for domain-incremental continual learning across diverse organs, diseases, and institutions. The method demonstrates strong performance, and robust experimental design, making it a valuable contribution to continual learning in medical imaging.

**Major Strengths**

Comprehensive and well-designed experimental setup: Includes various domain shifts, extensive comparisons, a small ablation study, and interpretability analysis.

Clear writing and strong motivation: The introduction effectively frames the problem and the paper is easy to follow.

Innovative method: Presents a novel adaptation of generative latent replay with attention mechanisms tailored to the medical imaging context. Further, the authors show applicability across different organs and institutions, highlighting real-world relevance.

**Weaknesses**

Attention map interpretability is included, but it is not a particularly reliable measure for explaining model decisions. Also, it is unclear what the artifacts are that are described in the text that should be contained in Figure 2. Fortunately, this has an easy fix: Clearly mark or annotate the "artifacts" in the attention at t=3/5 as referenced in the text, using arrows or circles for clarity.


In Table 3: "naive," "cumulative," and "joint" are never highlighted, while sometimes having the highest score. In the text it is noted that they serve as upper/lower bounds. However, it is not mentioned why they are not highlighted. Similar as before this has an easy fix and ideally state in the caption that these are not highlighted to being upper lower bounds or similar information about the highlighting.


This paper is a strong example of what makes a good paper: a well-justified, innovative approach to continual learning in medical imaging with great experimental evaluation.

---

### Official Review · Reviewer_Tkcy · 2025-07-14
**CL Review**

**Rating:** 3
**Confidence:** 4

**Review:**

The authors propose an approach for episodic continual learning in computational pathology (CPath) classification tasks. Their method leverages foundation model embeddings and employs a Gaussian Mixture Model (GMM) as a generative model to represent the patch embeddings from previous tasks. New patch embeddings from incoming tasks are used to update the GMM, thereby hypothesizing that continual learning can be achieved without the need for explicit buffering or storage of past data. This idea is both innovative and timely, particularly for CPath applications where models are expected to handle sequential tasks from heterogeneous datasets.
Strengths:
Innovative Idea: The use of a generative GMM to model past task embeddings in a continuous learning framework is a creative approach that addresses a key challenge in CPath: the need for models that can adapt to new tasks without catastrophic forgetting.
Relevance to CPath: Given the variability among datasets (e.g., CRC-TCGA, CRC-PAIP), this method promises potential benefits by allowing a unified model to sequentially learn across different tasks.
Practical Significance: With the increasing deployment of AI models in clinical settings, the ability to update models continually without storing large amounts of historical data is particularly attractive.
Areas for Improvement:
Baseline and SoTA Results: The paper would benefit from including baseline performance metrics of the original feature embeddings for each task as well as the current state of the art results for these tasks in the literature. For instance, more details on the results obtained when using the embeddings for each dataset (e.g., CRC-TCGA, CRC-PAIP) independently should be provided. This is especially important given that the reported AUROC values (such as those for MSI prediction in CRC) appear to be lower than what has been published in the literature (typically above 0.90). Providing these baselines would help contextualize the improvements (or lack thereof) achieved by the continuous learning approach.
Comparative Analysis: While the authors compare their method with other continual learning approaches, it is also essential to compare their results against the original baseline models that do not employ continual learning mechanisms (e.g. simple fine tuning). Moreover, the significant drop in performance (as measured by Backward Transfer, BWT) should be addressed. An in-depth discussion on the cause of the drop and potential ways to mitigate catastrophic forgetting would be beneficial.
Experimental Robustness: The reported results appear to be from a single experimental run. To ensure robustness and generalizability, the experiments should be repeated multiple times, with results averaged over several runs along with measures of variability (such as standard deviation or confidence intervals).
Data Considerations: There is a concern regarding the choice of data in the CPTAC dataset, where some images include frozen section slides rather than diagnostic slides. The paper should clearly indicate whether these differences have been controlled for or explicitly justified within the context of the study.
Presentation and Writeup: While the narrative is largely clear, the paper would significantly benefit from an algorithmic description or pseudo-code that outlines the continual learning process. This addition would help readers understand the step-by-step mechanics of using the GMM for generative replay and updating the model.
The core idea of using a Gaussian Mixture Model as a generative mechanism for episodic continuous learning in CPath is promising and addresses a critical need in the field. However, the paper requires substantial improvements in experimental analysis, comparative evaluation, and clarity in presentation. Addressing these points would help in establishing the efficacy of the proposed method and provide clearer insights for subsequent research in continual learning for computational pathology.